# Prism: Spectral-Aware Block-Sparse Attention

**Xinghao Wang** [1 2]  **Pengyu Wang** [1 2]  **Xiaoran Liu** [1 2 3]  **Fangxu Liu** [4]  **Jason Chu** [4]  **Kai Song** [4]  **Xipeng Qiu** [1 2 3]

## Abstract

Block-sparse attention is promising for accelerating long-context LLM pre-filling, yet identifying relevant blocks efficiently remains a bottleneck. Existing methods typically employ coarse-grained attention as a proxy for block importance estimation, but often resort to expensive token-level searching or scoring, resulting in significant selection overhead. In this work, we trace the inaccuracy of standard coarse-grained attention via mean pooling to a theoretical root cause: the interaction between mean pooling and Rotary Positional Embeddings (RoPE). We prove that mean pooling acts as a low-pass filter that induces destructive interference in high-frequency dimensions, effectively creating a "blind spot" for local positional information (e.g., slash patterns). To address this, we introduce Prism, a training-free spectral-aware approach that decomposes block selection into high-frequency and low-frequency branches. By applying energy-based temperature calibration, Prism restores the attenuated positional signals directly from pooled representations, enabling block importance estimation using purely block-level operations, thereby improving efficiency. Extensive evaluations confirm that Prism maintains accuracy parity with full attention while delivering up to $5.1\times$ speedup. Code available at https://github.com/xinghaow99/prism.

## 1. Introduction

The capacity to process extensive contexts is a defining characteristic of modern Large Language Models (LLMs), unlocking applications ranging from repository-level code understanding to hour-long video understanding (Bai et al., 2024; Wu et al., 2024). However, handling such long contexts is non-trivial, as the self-attention mechanism scales

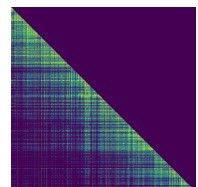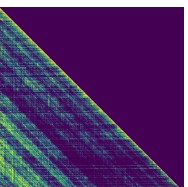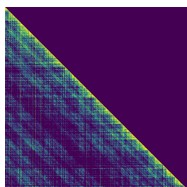

*Figure 1.* **Spectral Disentanglement of Attention Patterns.** We visualize the attention score matrices computed using different spectral bands of RoPE. **(Left) Low-Frequency Band:** Captures global *semantic dependencies* (e.g., block-sparse patterns / vertical lines), acting as the semantic backbone. **(Middle) High-Frequency Band:** Strictly encodes fine-grained *relative locality* (e.g., slash lines), which is critical for local coherence. **(Right) Full Spectrum:** The superposition of both patterns.

quadratically with sequence length (Vaswani et al., 2017), resulting in massive computational intensity during the token-parallel pre-filling phase and bottlenecking practical deployment. To mitigate this, block-sparse attention has emerged as a promising solution, approximating full attention by computing only a subset of relevant blocks. The efficacy of this approach hinges on *block importance estimation*: efficiently identifying relevant blocks without full computation. Standard training-free methods typically employ mean pooling (Jiang et al., 2024; Lai et al., 2025) as a coarse-grained proxy. However, this proxy is often inaccurate, forcing state-of-the-art methods to rely on expensive heuristic search and token-level verification to maintain performance. This creates a fundamental trade-off: the heavy estimation overhead often negates the sparsity gains, causing these methods to underperform highly optimized full attention implementations (e.g., FlashAttention (Dao et al., 2022)) at moderate sequence lengths.

In this work, we trace the inaccuracy of standard coarse-grained attention to a theoretical root cause: the spectral interaction between mean pooling and Rotary Positional Embeddings (RoPE) (Su et al., 2024). As illustrated in Figure 1, the spectral heterogeneity of RoPE naturally disentangles attention into distinct structural patterns: high-frequency dimensions strictly encode fine-grained relative positions, while low-frequency dimensions capture global semantic dependencies, manifesting as divergent sparse patterns. However, we mathematically prove that mean pooling acts as a **Low-Pass Filter**. In high-frequency dimensions, the rapid rotation of RoPE vectors induces **destructive interference**

---

[1]Fudan University [2]OpenMOSS Team [3]Shanghai Innovation Institute [4]ByteDance. Correspondence to: Xipeng Qiu <xpqiu@fudan.edu.cn>.

*Proceedings of the $43^{rd}$ International Conference on Machine Learning*, Seoul, South Korea. PMLR 306, 2026. Copyright 2026 by the author(s).

during aggregation, causing the signal magnitude to collapse. This phenomenon creates a spectral "Blind Spot" that effectively erases fine-grained positional information (e.g., slash patterns) from the pooled representation, explaining why standard methods struggle to maintain local coherence without expensive corrections.

To address this, we introduce **Prism**, a spectral-aware framework that disentangles block importance estimation into two parallel branches. Instead of treating embeddings as monolithic vectors, Prism explicitly separates the attenuated high-frequency band from the robust low-frequency band. By applying a novel energy-based temperature calibration, Prism restores the attenuated positional signals from pooled representations. This design enables Prism to perform precise importance estimation using **exclusively block-level operations**, eliminating the selection bottleneck common in prior works.

We evaluate Prism with diverse long-context capabilities, ranging from language modeling (PG19 (Rae et al., 2020)), long-context understanding (LongBench (Bai et al., 2024)), long-context retrieval (RULER (Hsieh et al., 2024)), and video understanding (VideoMME (Fu et al., 2025) & LongVideoBench (Wu et al., 2024)). Experiments demonstrate that Prism closely matches the accuracy of full attention while delivering substantial speedups compared to FlashAttention and state-of-the-art sparse attention methods. Our contributions are summarized as follows: (1) **Theoretical Insight:** We identify mean pooling as a low-pass filter under RoPE, revealing the "Blind Spot" responsible for the failure of standard block importance estimation. (2) **Methodology:** We propose Prism, a training-free framework utilizing dual-band scoring and energy-based calibration to explicitly preserve high-frequency positional information without token-level overhead. (3) **SOTA Efficiency:** Prism achieves state-of-the-art accuracy-speedup trade-offs, delivering up to $5.1\times$ speedup at 128K tokens while outperforming baselines in latency across all sequence lengths.

## 2. Related Work

**Block-Sparse Attention** The quadratic computational complexity of the self-attention mechanism (Vaswani et al., 2017) poses a significant bottleneck for processing long contexts in modern LLMs. Fortunately, as a result of the softmax operation, learned attention matrices often exhibit highly sparse patterns; that is, a small subset of tokens accounts for the majority of the attention mass, providing an opportunity to reduce computational overhead. Early sparse attention approaches relied on static sparse patterns, such as fixed sliding windows (Child et al., 2019), dilated windows (Beltagy et al., 2020), or global "sink" tokens (Xiao et al., 2024) to maintain local coherence and stability. However, static patterns often fail to capture long-range dependencies scattered arbitrarily across the sequence (the "needle in a haystack" problem). Consequently, recent research has shifted toward dynamic sparse attention, where the attention pattern is determined adaptively based on the input. To implement this efficiently on hardware, block-sparse approaches partition the sequence into fixed-size blocks (e.g., 128×128). This design naturally aligns with the tiling mechanism of FlashAttention (Dao et al., 2022), which decomposes computation into contiguous blocks for I/O awareness. By restricting the dense computation and online accumulation to a selected subset of block pairs, this granularity allows for optimized GPU kernels (e.g., via Triton or CUDA) while significantly reducing the number of FLOPs during the compute-bound pre-filling stage.

**Block Importance Estimation** The central challenge in dynamic block-sparse attention is *block importance estimation*: identifying which Key blocks are relevant to a given Query block without incurring the quadratic cost of the full attention matrix. In the scope of pre-filling, existing training-free approaches typically rely on coarse-grained proxies combined with heuristic pattern matching. Methods such as MInference (Jiang et al., 2024) and FlexPrefill (Lai et al., 2025) employ offline or online search strategies to classify attention heads into pre-defined categories (e.g., "Vertical Slash" or "Block-Sparse"). Consequently, they adopt divergent estimation techniques, utilizing coarse-level attention for semantic retrieval heads while falling back to selection against certain patterns. Other works aim for a unified estimation metric. SpargeAttention (Zhang et al., 2025) adopts coarse-level attention for all heads while enforcing blocks with low intra-block similarity. XAttention (Xu et al., 2025) introduces an antidiagonal scoring mechanism to capture both block-sparse and vertical-slash patterns, while PBS-Attn (Wang et al., 2025) utilizes token permutation to cluster critical tokens for better separability. However, these methods typically involve additional token-level operations, which significantly degrade block selection efficiency, particularly at moderate sequence lengths where the selection overhead outweighs the sparsity gains.

## 3. Method

### 3.1. Preliminaries

**Coarse-grained Attention** Block-sparse attention requires a block mask $\mathcal{M}$ to determine if a block pair $(u, v)$ should be computed. For efficient estimation of $\mathcal{M}$, a typical approach is to compute a coarse-grained attention matrix $\bar{\mathbf{S}}$. Formally, let $\mathbf{Q}, \mathbf{K}, \mathbf{V} \in \mathbb{R}^{L \times d}$ denote the query, key, and value matrices, where $L$ is the sequence length and $d$ is the head dimension. The sequence is partitioned into $N = \lceil \frac{L}{B} \rceil$ blocks, where $B$ is the block size. For the $u$-th query block and $v$-th key block, let $\mathcal{I}_u$ and $\mathcal{I}_v$ denote the

sets of token indices belonging to each block, respectively. Coarse-grained attention typically compresses each block into a single representative vector using mean pooling:

$$\bar{\mathbf{q}}_u = \frac{1}{B} \sum_{i \in \mathcal{I}_u} \mathbf{q}_i, \quad \bar{\mathbf{k}}_v = \frac{1}{B} \sum_{j \in \mathcal{I}_v} \mathbf{k}_j \qquad (1)$$

Let $\bar{\mathbf{Q}}, \bar{\mathbf{K}} \in \mathbb{R}^{N \times d}$ be the matrices formed by stacking these pooled vectors. Then the coarse-grained attention matrix is computed as:

$$\bar{\mathbf{S}} = \text{softmax}\left(\frac{\bar{\mathbf{Q}}\bar{\mathbf{K}}^\top}{\sqrt{d}}\right) \qquad (2)$$

Finally, a top-$k$ or top-$p$ selection is applied to $\bar{\mathbf{S}}$ to generate the binary mask $\mathcal{M} \in \{0, 1\}^{N \times N}$.

**Spectral Structure of RoPE** Modern large language models (LLMs) (Grattafiori et al., 2024; Yang et al., 2025; GLM-4.5 Team et al., 2025; Team Olmo et al., 2025) typically employ rotary positional embeddings (RoPE) (Su et al., 2024) to inject positional information. RoPE rotates feature pairs in the complex plane. Let $x_n^{(j)}$ denote the $j$-th feature pair of a vector at position $n$, represented as a complex number. The embedding is rotated by an angle dependent on the position $n$ and a frequency $\theta_j$:

$$\mathbf{x}_n^{(j)} = \mathbf{x}_{nope}^{(j)} \cdot e^{in\theta_j} \qquad (3)$$

Crucially, the rotation frequencies are defined as a geometric sequence decaying across the feature dimension index $j \in \{0, \ldots, d/2 - 1\}$:

$$\theta_j = b^{-2j/d} \qquad (4)$$

where $b$ is the base (e.g. 1M for Qwen3). This definition creates a **Spectral Heterogeneity** (Liu et al., 2024) across the embedding dimensions: (1) **High-Frequency Band** ($j \to 0$): Dimensions with low indices possess large $\theta_j$, resulting in rapid rotation. These dimensions encode fine-grained, relative positional information (e.g., local context). (2) **Low-Frequency Band** ($j \to d/2$): Dimensions with high indices possess $\theta_j \to 0$, resulting in negligible rotation over long distances. These dimensions behave similarly to absolute embeddings, primarily encoding global semantic content.

This spectral distribution implies that linear operations applied across the sequence dimension, such as the mean pooling defined in Eq. 1, will exhibit frequency-dependent behaviors, a phenomenon we analyze in the following section.

**Sparse Patterns of Attention** Extensive empirical analysis (Xiao et al., 2024; Jiang et al., 2024; Lai et al., 2025) reveals that attention matrices in pre-trained LLMs are not

uniformly sparse but exhibit distinct structural characteristics, most notably the vertical slash patterns and block-sparse patterns. Prior works typically treat these patterns as mutually exclusive properties of specific attention heads, employing heuristic classifiers to assign distinct estimation strategies (Jiang et al., 2024; Lai et al., 2025). Although Xu et al. (2025) attempted to capture both patterns via a unified antidiagonal scoring mechanism, their approach still incurs additional token-level operations, resulting in significant selection overhead at long sequence lengths. We challenge this head-level dichotomy. We posit that these patterns are not spatially separated across heads but are instead **spectrally disentangled within individual heads**.

As visualized in Figure 1, the high-frequency spectral bands of RoPE (low indices) strictly encode relative locality (slash patterns), while the low-frequency bands (high indices) capture global semantic dependencies (block-sparse patterns). This spectral observation motivates our frequency-decomposed approach.

### 3.2. Mean Pooling as a Low-Pass Filter

To facilitate efficient block importance estimation, mean pooling(Eq. 1) serves as a common technique to compress a block into a single representative vector. In this section, we theoretically analyze the impact of mean pooling with the consideration of RoPE, which explains why existing methods had to resort to token-level operations for accurate block importance estimation.

**Geometric Summation of Mean Pooling** Consider the $j$-th frequency pair of the query vector. Under RoPE, the embedding at position $n$ can be decomposed into a content component $c^{(j)}$ and a positional rotation $e^{in\theta_j}$. Assuming the semantic content $c^{(j)}$ remains relatively stable within the local context of a block (a standard assumption for adopting mean pooling), applying the mean pooling over a block of size $B$ starting at position $n_0$ can be formulated as a geometric series summation:

$$\bar{\mathbf{q}}^{(j)} \approx \frac{c^{(j)}}{B} \sum_{k=0}^{B-1} e^{i(n_0+k)\theta_j} = \frac{c^{(j)} e^{in_0\theta_j}}{B} \underbrace{\left(\sum_{k=0}^{B-1} e^{ik\theta_j}\right)}_{\text{Geometric Sum}} \qquad (5)$$

**Spectral Attenuation** The magnitude of this pooled vector dictates the signal strength available for dot-product retrieval. By evaluating the geometric sum, we derive the **Spectral Attenuation Factor** $\lambda_j(B)$, defined as the ratio of the pooled vector's magnitude to the original vector's magnitude:

$$\lambda_j(B) \triangleq \frac{|\bar{\mathbf{q}}_j|}{|c|} = \left|\frac{1}{B} \sum_{n=0}^{B-1} e^{in\theta_j}\right| = \frac{1}{B}\left|\frac{\sin(B\theta_j/2)}{\sin(\theta_j/2)}\right| \qquad (6)$$

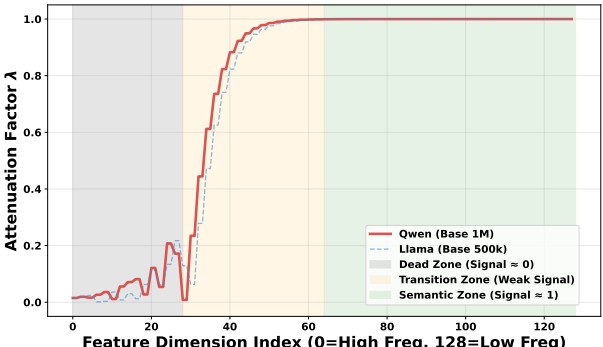

*Figure 2.* Spectral attenuation factor $\lambda_j(B)$ with block size $B = 128$ and head dimension $d = 128$.

For small frequencies, this function converges to the normalized sinc function:

$$\lambda_j(B) \approx \left| \text{sinc}\left( \frac{B\theta_j}{2\pi} \right) \right| \tag{7}$$

A detailed derivation is provided in Appendix A. This derivation mathematically reveals that mean pooling functions as a **Low-Pass Filter**: (1) **Destructive Interference** ($\lambda_j \to 0$): In the high-frequency band where the block size covers full rotation periods ($B\theta_j \approx 2\pi k$), the vectors sum to near-zero. For a standard block size $B = 128$, this creates a "Blind Spot" in the first $\approx 30$ dimensions (for Base 1M), effectively erasing local positional structures. (2) **Constructive Interference** ($\lambda_j \to 1$): In the low-frequency band where $\theta_j \to 0$, the rotations are negligible, and the signal magnitude is fully preserved.

We quantify this effect using a standard setting with block size $B = 128$ and head dimension $d = 128$, considering RoPE bases $b = 10^6$ (Qwen3) and $b = 5 \times 10^5$ (LLaMa 3.1), as visualized in Figure 2. Taking Qwen3 as an example, destructive interference reaches its peak ($\lambda_j \approx 0$) when the total rotation $B\theta_j = 2\pi$. We solve for the corresponding feature dimension index $2j$:

$$B \cdot b^{-2j/d} = 2\pi \implies 2j = d \cdot \frac{\ln(B/2\pi)}{\ln b} \tag{8}$$

Substituting the values yields a cutoff dimension of $2j \approx 28$. Based on this derivation, the spectrum in Figure 2 divides into three distinct regimes: (1) **The Dead Zone** ($0 \leq 2j \lesssim 30$): The signal magnitude is effectively zero due to full phase cancellation. (2) **The Transition Zone** ($30 \lesssim 2j \lesssim 60$): The signal begins to recover but remains heavily attenuated ($\lambda < 1$). (3) **The Semantic Zone** ($2j > 60$): The signal magnitude is fully preserved, capturing global semantic information.

This analysis theoretically justifies why standard coarse-grained attention is "blind" to fine-grained positional structures encoded in the high-frequency band.

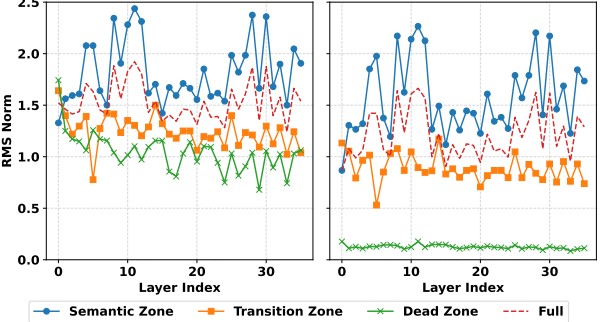

*Figure 3.* Comparison of Query RMS norms before and after pooling. **Left (Token-level):** While the Semantic Zone (blue) holds the highest energy, the Dead Zone (green) maintains a **robust magnitude** (RMS $\approx 1.0$), confirming that high-frequency dimensions are actively utilized by the pre-trained model. **Right (Block-pooled):** After pooling, energy in the Dead Zone **collapses to near-zero** due to destructive interference, while the Semantic Zone preserves its magnitude.

### 3.3. Energy Analysis

To verify whether the theoretical attenuation derived in Section 3.2 manifests in actual model representations, we analyze the spectral energy distribution using Qwen3-8B. We measure the RMS norms of the query vectors before and after mean pooling across the three spectral zones defined in Figure 2. Ideally, if pooling were lossless, the block-level RMS should mirror the token-level RMS. However, Figure 3 reveals a distinct **Spectral Divergence**: At the token level (Left), the Dead Zone maintains robust magnitude (RMS $\approx 1.0$), confirming that high-frequency positional features are intrinsically significant to the pre-trained model. In contrast, the block-pooled representation (Right) exhibits a dramatic **Energy Collapse** in the Dead Zone (RMS $\approx 0.1$), empirically validating that mean pooling acts as a low-pass filter that suppresses local positional information. Crucially, the RMS of the Semantic Zone consistently surpasses the Full spectrum. This intrinsic divergence is **significantly exacerbated** post-pooling, as the Full vector is further diluted by the "dead weight" of attenuated high-frequency dimensions. This widened energy gap necessitates the frequency-dependent calibration proposed next.

### 3.4. Prism: Spectral-Aware Block Selection

To resolve the spectral bias identified above, we propose **Prism**, a framework that decomposes block selection into two spectral branches based on their characteristics. The overall procedure is summarized in Figure 4 and consists of two core components: (1) **Dual-Band Block Importance Estimation**, which explicitly isolates the high-frequency and low-frequency bands to avoid signal interference during aggregation; and (2) **Energy-Based Temperature Calibration**, which derives branch-specific temperatures from spectral energy distributions, restores the logit magnitudes with-

out any hyperparameter tuning. Crucially, this design enables Prism to perform estimation using exclusively **block-level operations**, minimizing selection overhead.

**Dual-Band Block Importance Estimation**   To best preserve information from both spectral bands, we propose a dual-band block importance estimation strategy that avoids interference between the two bands.

Let $\mathbf{Q}, \mathbf{K} \in \mathbb{R}^{L \times d}$ denote the input query and key matrices. We explicitly isolate the High-Frequency Band by slicing the first $d_{high}$ dimensions, yielding $\mathbf{Q}_{high}, \mathbf{K}_{high} \in \mathbb{R}^{L \times d_{high}}$. Similarly, we slice the last $d_{low}$ dimensions to form the Low-Frequency Band, $\mathbf{Q}_{low}, \mathbf{K}_{low} \in \mathbb{R}^{L \times d_{low}}$. Subsequently, mean pooling with block size $B$ is applied to the high-frequency and low-frequency bands independently, obtaining $\bar{\mathbf{Q}}_{high}, \bar{\mathbf{K}}_{high} \in \mathbb{R}^{N \times d_{high}}$ and $\bar{\mathbf{Q}}_{low}, \bar{\mathbf{K}}_{low} \in \mathbb{R}^{N \times d_{low}}$, where $N = \lceil \frac{L}{B} \rceil$. With the pooled representations, we compute the coarse-grained importance scores for each spectral band $z \in \{\text{high}, \text{low}\}$. Furthermore, to account for the distinct spectral energy densities caused by attenuation (as observed in Figure 3), we introduce branch-specific temperature scaling factors $\tau_{high}$ and $\tau_{low}$:

$$\bar{\mathbf{S}}_z = \text{softmax}\left( \frac{\bar{\mathbf{Q}}_z \bar{\mathbf{K}}_z^\top}{\tau_z \sqrt{d_z}} \right), \quad \text{for } z \in \{\text{high}, \text{low}\} \quad (9)$$

Based on the probability distributions $\bar{\mathbf{S}}_{high}$ and $\bar{\mathbf{S}}_{low}$, we generate binary block masks $\mathcal{M}_{high}$ and $\mathcal{M}_{low}$ by selecting the top-$p$ cumulative probability mass for each query block. The final block-sparse mask $\mathcal{M}$ is obtained by the union of these branch-specific selections:

$$\mathcal{M} = \mathcal{M}_{high} \cup \mathcal{M}_{low} \quad (10)$$

**Energy-Based Temperature Calibration**   To align the logit magnitude of the individual spectral bands to the scale of the full spectrum, we derive the branch-specific temperatures $\tau_z$ based on the spectral energy distribution. We employ RMS norm to represent the spectral energy density of a pooled matrix $\bar{\mathbf{X}} \in \mathbb{R}^{N \times d}$, where $\text{RMS}(\bar{\mathbf{X}}) = \sqrt{\frac{1}{N} \sum_{u=1}^{N} \frac{\|\bar{\mathbf{x}}_u\|^2}{d}}$. Consider attention logits $L_{full} = (\bar{\mathbf{Q}}_{full} \bar{\mathbf{K}}_{full}^\top)/\sqrt{d}$. Since the dot product accumulates magnitude across $d$ dimensions, the scale of these logits follows:

$$|L_{full}| \propto \sqrt{d} \cdot \text{RMS}(\bar{\mathbf{Q}}_{full}) \text{RMS}(\bar{\mathbf{K}}_{full}) \quad (11)$$

Similarly, for a spectral branch $z$ using subspace dimension $d_z$, the uncalibrated logits $L_z$ scale as:

$$|L_z| \propto \sqrt{d_z} \cdot \text{RMS}(\bar{\mathbf{Q}}_z) \text{RMS}(\bar{\mathbf{K}}_z) \quad (12)$$

To restore the signal strength of the partial branch to the baseline level (i.e., $|L_z|/\tau_z \approx |L_{full}|$), we derive the cali-

```python
def prism(Q, K, d_h, d_l, B, p):
    # Setup dimensions
    bs, h, L, d = Q.shape
    N = L // B

    # 1. Pooling & Slicing
    Qb, Kb = pool(Q, B), pool(K, B)
    Qh, Ql = Qb[..., :d_h], Qb[..., -d_l:]
    Kh, Kl = Kb[..., :d_h], Kb[..., -d_l:]

    # 2. RMS Calculation
    rq, rk = rms(Qb), rms(Kb)
    rq_h, rk_h = rms(Qh), rms(Kh)
    rq_l, rk_l = rms(Ql), rms(Kl)

    # 3. Calibration (Eq. 13)
    th = sqrt(d_h/d) * (rq_h/rq) * (rk_h/rk)
    tl = sqrt(d_l/d) * (rq_l/rq) * (rk_l/rk)

    # 4. Dual-Band Scoring
    scale_h = sqrt(d_h) * th
    scale_l = sqrt(d_l) * tl
    logits = empty(bs, h, 2N, N)
    logits[..., :N, :]=(Qh @ Kh.T) / scale_h
    logits[..., N:, :]=(Ql @ Kl.T) / scale_l

    # 5. Selection
    P = softmax(logits, dim=-1)
    Mh, Ml = top_p(P, p).split(N, dim=-2)

    return Mh | Ml
```

*Figure 4.* PyTorch-style implementation of Prism. Prism exclusively uses block-level operations for best efficiency. See Appendix C for `top_p` implementation.

bration factor:

$$\tau_z \approx \sqrt{\frac{d_z}{d}} \cdot \frac{\text{RMS}(\bar{\mathbf{Q}}_z)}{\text{RMS}(\bar{\mathbf{Q}}_{full})} \cdot \frac{\text{RMS}(\bar{\mathbf{K}}_z)}{\text{RMS}(\bar{\mathbf{K}}_{full})} \quad (13)$$

## 4. Experiments

### 4.1. Setup

**Benchmarks, Models & Baselines**   To evaluate the versatility and robustness of Prism, we conduct experiments across five categories of long-context tasks: (1) **Language Modeling** using PG19 (Rae et al., 2020); (2) **Long-Context Understanding** using LongBench (Bai et al., 2024); (3) **Long-Context Retrieval** using RULER (Hsieh et al., 2024); (4) **Video Understanding** using VideoMME (Fu et al., 2025) and LongVideoBench (Wu et al., 2024); and (5) **Video Generation** using HunyuanVideo (Kong et al., 2024) evaluated with VBench prompts (Huang et al., 2024). We employ state-of-the-art models including **Llama-3.1-8B-Instruct** (128K) (Grattafiori et al., 2024) and the **Qwen3-8B** (Yang et al., 2025). Notably, for Qwen3-8B, we apply **YaRN** (Peng et al., 2024) extrapolation to extend the context from 32K to 128K. For multimodal tasks, we utilize

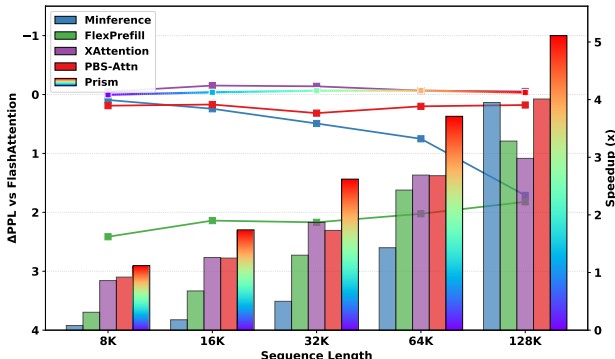

*Figure 5.* **Language modeling performance on PG19.** We compare the Perplexity Degradation (ΔPPL, solid lines, left axis) and Speedup (bars, right axis) across sequence lengths. Prism achieves a **double win**: it shows no perplexity degradation (sticking to the $\Delta \approx 0$ line) while delivering the highest speedup ($5.1\times$ at 128K), significantly outperforming baselines that trade off accuracy for speed or suffer from high selection overhead.

**Qwen3-VL-8B** (Bai et al., 2025). This selection specifically enables us to verify Prism's generalization to RoPE variants, including YaRN, Interleaved M-RoPE, and 3D-RoPE. We compare Prism with **FlashAttention-2** (Dao, 2024) (full attention baseline), and state-of-the-art training-free dynamic block-sparse methods: **MInference** (Jiang et al., 2024), **FlexPrefill** (Lai et al., 2025), **XAttention** (Xu et al., 2025), and **PBS-Attn** (Wang et al., 2025). To ensure fair comparison, we use the official recommended configurations for all baselines. Details in Appendix D.

**Implementation Details** For Prism, we use a block size $B = 128$ based on the trade-off analysis in Appendix F. Guided by the spectral analysis in Figure 2, we configure the spectral bands as $d_{high} = 64$ and $d_{low} = 96$. This configuration ensures robust signal coverage by overlapping the transition zone, while strictly aligning dimension sizes with multiples of 32 to maximize Tensor Core throughput on GPUs. For Top-P selection, we use a threshold $p = 0.95$ for Llama-3.1-8B-Instruct and $p = 0.93$ for Qwen models to balance the trade-off between efficiency and accuracy. For importance estimation and block-sparse attention, we implement custom Triton kernels for best efficiency.

### 4.2. Main Results

**Language Modeling** We evaluate the modeling capability on long-context sequences using the PG19 benchmark. Figure 5 visualizes the scalability of Prism compared to baselines, plotting Perplexity Degradation (ΔPPL) and Speedup. Notably, Prism demonstrates superior robustness, maintaining a perplexity virtually identical to the Full Attention baseline (ΔPPL $\approx 0$) across all context lengths. In contrast, baselines like MInference and FlexPrefill suffer from significant perplexity degradation as sequence length increases,

especially at 128K. While XAttention achieves high fidelity comparable to Prism, it is bottlenecked by significant estimation overhead. This becomes critical at extreme lengths: at 128K, XAttention is limited to a $3.0\times$ speedup, whereas Prism achieves $5.1\times$. Consequently, Prism achieves a **double win**, delivering the highest speedup while simultaneously maintaining the perplexity of full attention.

**Long-Context Understanding** Table 1 presents the evaluation results on LongBench. Prism demonstrates exceptional robustness, achieving average scores of 41.08 on Llama-3.1-8B-Instruct and 39.12 on Qwen-3-8B, showing negligible degradation ($< 0.4\%$) compared to the full attention baseline. While MInference achieves similar accuracy, it relies on a fixed budget strategy that, at the moderate sequence lengths of LongBench ($< 16K$), often results in selecting nearly all tokens. Consequently, it degenerates to full attention while incurring additional estimation overhead, failing to provide meaningful sparsity. In contrast to other sparse baselines, Prism significantly outperforms FlexPrefill and XAttention on average for both models. Notably, Prism even slightly outperforms full attention on specific tasks (e.g., 58.36 vs. 56.69 on Qwen-3 Few-shot). We attribute this gain to the explicit preservation of high-frequency positional signals. By recovering the fine-grained relative structure essential for Induction Heads (Olsson et al., 2022), Prism enhances the model's ability to perform in-context pattern copying. Furthermore, unlike full attention, Prism filters out irrelevant semantic blocks, effectively denoising the context for these position-sensitive heads.

**Long-Context Retrieval** Table 2 reports the evaluation results on RULER. As shown in the table, all methods show comparable performance with their configured threshold parameters. However, it is crucial to note that Prism achieves this parity using exclusively block-level operations in semantic retrieval. In contrast, baselines like MInference and FlexPrefill rely on token-level estimation using the last query block, a heuristic that is inherently advantageous for RULER's format, where the query is typically positioned at the end. Despite not being explicitly optimized for such structure, Prism's Low-Frequency Branch successfully handles these retrieval tasks, validating that our spectral calibration preserves sufficient semantic recall. Notably, the robust results on the YaRN-extrapolated Qwen3-8B demonstrate Prism's generalizability to RoPE variants without requiring additional adaptations; we provide the corresponding zone-wise analysis in Appendix B.

**Video Understanding** To assess the generalizability of Prism to multimodal scenarios, we evaluate performance on VideoMME and LongVideoBench using Qwen3-VL-8B. As shown in Table 3, Prism outperforms existing approaches on both benchmarks, achieving performance comparable to

*Table 1.* **Performance comparison on LongBench.**

| Method | Single-Doc QA | Multi-Doc QA | Summarization | Few-shot Learning | Code | Synthetic | Avg. |
|---|---|---|---|---|---|---|---|
| *Llama-3.1-8B* | | | | | | | |
| Full | 47.51 | 43.28 | 25.9 | 45.92 | 18.01 | 68.18 | 41.47 |
| MInference | **47.42** | **42.54** | 25.85 | 45.58 | 17.84 | **67.6** | **41.14** |
| FlexPrefill | 46.13 | 41.49 | 25.85 | **46.63** | 17.68 | 25.61 | 33.90 |
| XAttention | 45.89 | 41.56 | **26.18** | 45.86 | 19.24 | 59.32 | 39.68 |
| PBS-Attn | 46.53 | 41.97 | 25.88 | 45.92 | **20.23** | 65.08 | 40.94 |
| **Prism** | 47.09 | 42.13 | 26 | 46.4 | 18.72 | 66.15 | 41.08 |
| *Qwen-3-8B* | | | | | | | |
| Full | 47.1 | 40.45 | 24.07 | 56.69 | 1.65 | 67 | 39.49 |
| MInference | **46.9** | **40.39** | 24.07 | 55.74 | 1.61 | **66.33** | **39.18** |
| FlexPrefill | 43.77 | 39.31 | 23.99 | 57.33 | 1.87 | 50.5 | 36.13 |
| XAttention | 44.49 | 40.09 | **24.12** | 57.27 | 1.29 | 65.67 | 38.82 |
| PBS-Attn | 44.83 | 40.02 | 24.04 | 56.74 | **2.58** | 65.83 | 39.01 |
| **Prism** | 46.47 | 40.08 | 24.01 | **58.36** | 1.64 | 64.17 | 39.12 |

*Table 2.* **Performance comparison on RULER.**

| Method | 4K | 8K | 16K | 32K | 64K | 128K | Avg. |
|---|---|---|---|---|---|---|---|
| *Llama-3.1-8B* | | | | | | | |
| Full | 95.42 | 94.38 | 93.38 | 87.98 | 84.72 | 77.77 | 88.94 |
| MInference | 95.43 | 94.46 | **93.42** | 87.22 | 83.07 | 71.04 | 87.44 |
| FlexPrefill | 93.8 | 92.44 | 93.28 | 87.92 | **84.74** | 72.41 | 87.43 |
| XAttention | 95.17 | 94.3 | 93.28 | **89.06** | 82.31 | 70.52 | 87.44 |
| PBS-Attn | **95.45** | 94.01 | 92.54 | 85.77 | 83.03 | 71.69 | 87.08 |
| **Prism** | 95.28 | **94.47** | 92.48 | 87.67 | 82.59 | **72.75** | **87.54** |
| *Qwen-3-8B(YaRN)* | | | | | | | |
| Full | 95.01 | 92.35 | 90.04 | 87.24 | 79.93 | 75.09 | 86.61 |
| MInference | **95.08** | **92.37** | **89.67** | 86.01 | 76.53 | 70.36 | 85.00 |
| FlexPrefill | 90.89 | 87.61 | 87.82 | 85.58 | 78.27 | **73.42** | 83.93 |
| XAttention | 94.55 | 91.03 | 87.91 | 84.37 | 77.73 | 72.01 | 84.60 |
| PBS-Attn | 94.83 | 92.1 | 88.18 | 85.97 | 78.41 | 72.03 | 85.25 |
| **Prism** | 94.84 | 90.95 | 87.69 | **86.88** | **78.58** | 72.65 | **85.27** |

*Table 3.* **Performance comparison on long video understanding tasks with Qwen3-VL-8B.**

| Method | VideoMME | | | | LVB |
|---|---|---|---|---|---|
| | Short | Med. | Long | Overall | Overall |
| Full | 79.89 | 70.67 | 63.11 | 71.22 | 65.00 |
| MInference | 79.44 | 70.00 | 62.44 | 70.63 | 61.48 |
| FlexPrefill | 77.67 | **70.67** | 62.67 | 70.34 | 64.10 |
| XAttention | 79.22 | 69.78 | 63.44 | 70.81 | **64.25** |
| PBS-Attn | **79.56** | 69.56 | 62.89 | 70.67 | 64.17 |
| **Prism** | 79.00 | **70.67** | **64.00** | **71.22** | **64.25** |

*Table 4.* **Performance comparison on video generation.**

| Method | Threshold | PSNR↑ | SSIM↑ | LPIPS↓ | Speedup↑ |
|---|---|---|---|---|---|
| XAttention | 0.90 | 21.4 | 0.725 | 0.228 | 1.54× |
| | 0.95 | 23.3 | 0.797 | 0.171 | 1.33× |
| Prism | 0.90 | 20.7 | 0.713 | 0.259 | 1.76× |
| | 0.93 | 21.6 | 0.748 | 0.224 | 1.60× |
| | 0.95 | 22.4 | 0.775 | 0.198 | 1.50× |
| | 0.97 | 23.5 | 0.809 | 0.165 | 1.37× |

**Video Generation** We further evaluate Prism on dense video generation with HunyuanVideo, whose 3D-RoPE introduces rotations along temporal and spatial axes. We select Prism's spectral dimensions axis-wise according to the same attenuation criterion used for 1D RoPE (Eq. 8); the detailed configuration is provided in Appendix D.2. Table 4 reports fidelity to the full-attention output and end-to-end speedup under different sparsity thresholds. At comparable quality, Prism consistently achieves better efficiency than XAttention. For example, Prism with threshold 0.93 improves PSNR/SSIM/LPIPS over XAttention with threshold 0.90 (21.6/0.748/0.224 vs. 21.4/0.725/0.228) while increasing speedup from 1.54× to 1.60×. In the higher-quality regime, Prism with threshold 0.97 also slightly improves fidelity over XAttention with threshold 0.95 while yielding a higher speedup (1.37× vs. 1.33×). Qualitative results are provided in Appendix E. These results demonstrate that Prism's spectral-aware block selection extends beyond autoregressive pre-filling and long-video understanding to dense prediction workloads.

### 4.3. Efficiency Results

**Latency Comparison** We evaluate the attention prefilling latency and speedup of Prism compared to FlashAttention-2 and state-of-the-art sparse attention methods. Figure 6 illustrates the results across sequence lengths from 8K to 128K. Notably, Prism achieves consistent speedups across **all** sequence lengths. In contrast, baselines such as MInference and FlexPrefill only begin to outperform FlashAttention at 64K and 32K, respectively, as their

the full attention baseline. Crucially, in the *Long* split of VideoMME, where video durations range from 30 minutes to 1 hour (spanning 54K to 107K tokens), Prism surpasses the full attention baseline (64.00 vs. 63.11). We attribute this to the denoising effect of sparse attention, which effectively filters out irrelevant visual tokens, allowing the model to focus on the most salient visual information. These results also confirm the generalization of Prism to other multimodal RoPE variants (i.e., Interleaved M-RoPE (Bai et al., 2025)), demonstrating its robustness.

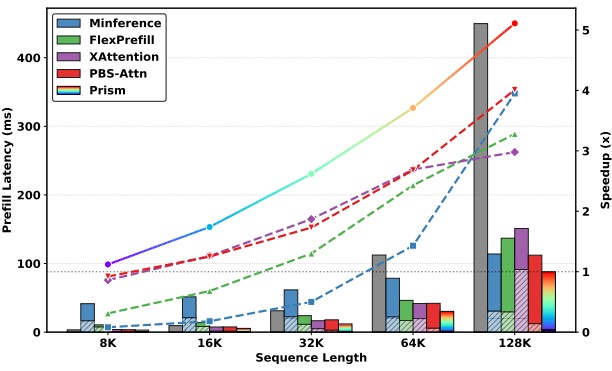

Figure 6. **Efficiency comparison on Llama-3.1-8B-Instruct with an H100 GPU.** We report pre-filling latency (bars, left axis) and speedup relative to FlashAttention-2 (lines, right axis). Shaded areas represent the block importance estimation time.

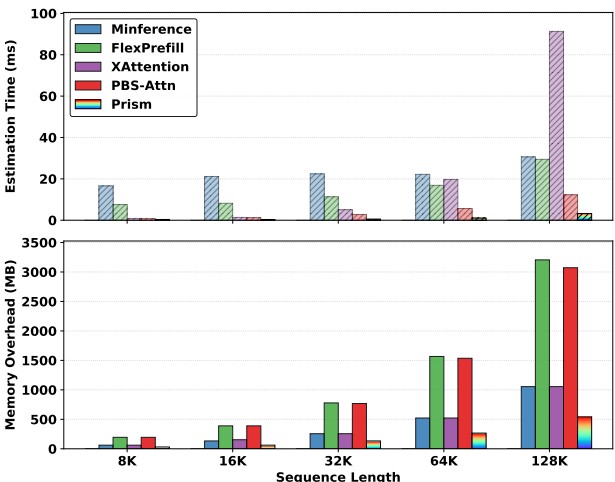

Figure 7. **Estimation overhead comparison.** The upper and lower panels illustrate the time and memory overhead of block importance estimation, respectively.

significant estimation overhead outweighs the sparsity gains at shorter lengths. While XAttention exhibits comparable speedups at moderate lengths, it suffers from diminishing returns at extreme lengths (e.g., 128K) due to increasing selection costs. Prism, however, preserves a robust speedup trajectory throughout, reaching **5.1×** at 128K.

**Estimation Overhead Comparison**  We further break down the estimation overhead in Figure 7. The results highlight the structural advantage of Prism's purely block-level design. Notably, Prism achieves the lowest estimation latency across all sequence lengths. Baselines like MInference and FlexPrefill maintain a relatively high constant overhead due to their token-level estimation components. Furthermore, XAttention suffers from a dramatic latency spike on long sequences ($\sim 85$ ms at 128K), primarily due to cost of its token-level access and computation. In contrast, Prism scales gracefully with sequence length, directly benefiting

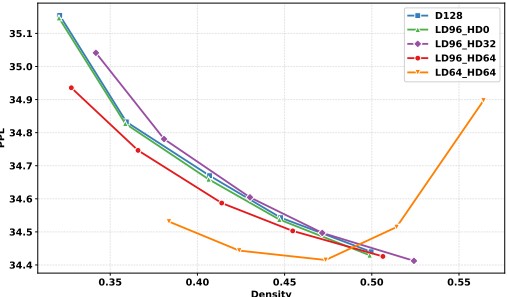

Figure 8. **Perplexity vs. Density with various dimension division strategies at 32K length.**

from its efficient matrix-multiplication-based scoring. This advantage extends to memory consumption, where Prism scales efficiently, requiring only $\sim 20\%$ of the memory used by FlexPrefill at 128K and remaining the lowest across all sequence lengths.

### 4.4. Ablation Studies

**Spectral Division**  We analyze the impact of different spectral band configurations on the Perplexity-Density trade-off in Figure 8 with the following findings: (1) **Mean Pooling is indeed a Low-Pass Filter**: Using only the low-frequency band (i.e., $d_{\text{low}} = 96$, $d_{\text{high}} = 0$) exhibits a nearly identical behavior to directly using the full dimension, even lower than the full dimension case, indicating that high-frequency components are acting only as noise in mean pooling block importance estimation. (2) **Necessity of Transition Zone in High-Frequency Band**: Restricting the high-frequency band to the theoretical dead zone ($d_{\text{high}} = 32$) yields suboptimal performance. This confirms that within the dead zone, positional signals are effectively erased by destructive interference. Consequently, attempting to align and calibrate this subspace only amplifies background noise, causing severe performance degradation. Extending the branch to $d_{\text{high}} = 64$ is thus critical to capture the recovering signals in the transition zone for effective restoration. (3) **Robustness of Overlapping**: While the aggressive semantic slicing ($d_{\text{low}} = 64$) appears promising at low densities, it exhibits performance instability (a U-shaped curve) at higher densities. We attribute this to the exclusion of the transition zone ($d \in [32, 64]$). By extending to $d_{\text{high}} = 96$ (red), we create a spectral overlap where the transition zone is covered by both branches. This design is crucial because the transition band, having moderate energy, acts as a **spectral regularizer** for the low-frequency branch: it moderates the energy density to prevent over-calibrated temperatures while ensuring signal continuity between positional and semantic regimes.

**Effect of Energy-Based Temperature Calibration**  We validate the necessity of our derived calibration formula by

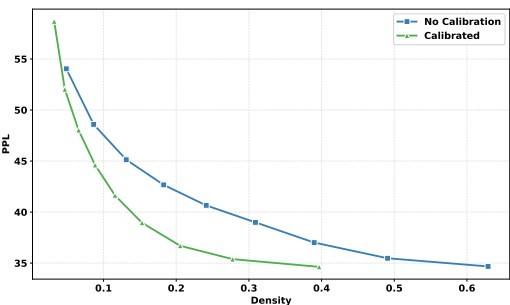

*Figure 9.* **Effect of Energy-Based Temperature Calibration.**

comparing the PPL-Density trade-off against a baseline with fixed temperature ($\tau_{\text{low}} = \tau_{\text{high}} = 1.0$). As shown in Figure 9, the calibrated configuration consistently dominates the uncalibrated one, pushing the Pareto frontier significantly towards better efficiency. Without calibration, the high-frequency logits remain attenuated, resulting in a flattened softmax distribution (high entropy). Consequently, the adaptive Top-$P$ policy fails to distinguish weak positional signals from background noise, forcing it to select a large number of irrelevant blocks, leading to an inefficient density inflation. In contrast, our calibration restores the logit magnitude, effectively sharpening the distribution to capture salient information within a limited density budget.

## 5. Conclusion

In this work, we identified the spectral attenuation induced by mean pooling under RoPE as the theoretical bottleneck for efficient block importance estimation. To address this, we introduced **Prism**, a training-free framework that explicitly preserves high-frequency information via dual-band scoring and energy-based calibration. By enabling precise selection using exclusively block-level operations, Prism achieves a $5\times$ speedup at 128K context while maintaining performance parity with full attention, offering a robust and scalable solution for long-context and multimodal LLMs.

## Impact Statement

This paper presents work whose goal is to advance the field of Machine Learning, specifically focusing on the efficiency of Large Language Models. By significantly reducing the computational overhead of long-context pre-filling, our method contributes to reducing the energy footprint of AI deployment. Furthermore, by lowering the hardware barrier for processing long sequences, this work helps democratize access to advanced LLM capabilities for researchers and practitioners with limited resources.

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

# A. Derivation of Spectral Attenuation Factor

In this section, we provide the detailed derivation of the spectral attenuation factor $\lambda_j(B)$ introduced in Eq. 6 and its convergence to the sinc function in Eq. 7.

## A.1. Setup and Geometric Summation

Consider the $j$-th frequency component of the query vector under Rotary Positional Embeddings (RoPE). We model the embedding at position $n$ as a complex number:

$$\mathbf{q}_n^{(j)} = c^{(j)} \cdot e^{in\theta_j} \tag{14}$$

where $c^{(j)}$ represents the semantic content (magnitude and initial phase) and $\theta_j$ is the rotation frequency. To isolate the effect of pooling on positional information, we assume the semantic content $c^{(j)}$ is locally stationary (constant) within the pooling window.

The mean pooling operation over a block of size $B$ (indexed locally from $k = 0$ to $B - 1$) yields the pooled vector $\bar{\mathbf{q}}^{(j)}$:

$$\bar{\mathbf{q}}^{(j)} = \frac{1}{B} \sum_{k=0}^{B-1} c^{(j)} \cdot e^{i(n_0+k)\theta_j} = \frac{c^{(j)} e^{in_0\theta_j}}{B} \sum_{k=0}^{B-1} e^{ik\theta_j} \tag{15}$$

where $n_0$ is the start position of the block. The term $S = \sum_{k=0}^{B-1} (e^{i\theta_j})^k$ is a geometric series with ratio $r = e^{i\theta_j}$. Applying the summation formula for a finite geometric series:

$$S = \frac{1 - (e^{i\theta_j})^B}{1 - e^{i\theta_j}} = \frac{1 - e^{iB\theta_j}}{1 - e^{i\theta_j}} \tag{16}$$

## A.2. Magnitude Calculation (The Dirichlet Kernel)

We define the attenuation factor $\lambda_j(B)$ as the ratio of the magnitude of the pooled vector to the magnitude of the original content $|c^{(j)}|$. Note that the phase term $|e^{in_0\theta_j}| = 1$ and thus does not affect the magnitude.

$$\lambda_j(B) \triangleq \frac{|\bar{\mathbf{q}}^{(j)}|}{|c^{(j)}|} = \frac{1}{B}|S| = \frac{1}{B} \left| \frac{1 - e^{iB\theta_j}}{1 - e^{i\theta_j}} \right| \tag{17}$$

To simplify the magnitude of the complex fraction, we utilize the half-angle identity $|1 - e^{i\phi}| = |e^{i\phi/2}(e^{-i\phi/2} - e^{i\phi/2})| = |-2i\sin(\phi/2)| = 2|\sin(\phi/2)|$. Applying this to both the numerator ($\phi = B\theta_j$) and the denominator ($\phi = \theta_j$):

$$\lambda_j(B) = \frac{1}{B} \frac{2|\sin(B\theta_j/2)|}{2|\sin(\theta_j/2)|} = \frac{1}{B} \left| \frac{\sin(B\theta_j/2)}{\sin(\theta_j/2)} \right| \tag{18}$$

This function is known as the normalized *Dirichlet kernel*, which describes the diffraction pattern of a discrete periodic lattice.

## A.3. Sinc Approximation

The RoPE frequencies are defined as $\theta_j = b^{-2j/d}$. For dimensions $j$ away from 0, the frequency $\theta_j$ decays exponentially and becomes very small ($\theta_j \ll 1$). We apply the small-angle approximation $\sin(x) \approx x$ to the denominator term[1]:

$$\sin(\theta_j/2) \approx \frac{\theta_j}{2} \tag{19}$$

Substituting this into the expression for $\lambda_j(B)$:

$$\lambda_j(B) \approx \frac{1}{B} \left| \frac{\sin(B\theta_j/2)}{\theta_j/2} \right| \tag{20}$$

---

[1]The small-angle approximation $\sin(x) \approx x$ holds due to the exponential decay of RoPE frequencies $\theta_j = b^{-2j/d}$. Taking Qwen3 ($b = 10^6, d = 128$) as an instance, the frequency drops to $\theta_{10} \approx 0.11$ by the 10th dimension pair. At this point, the relative error is already $< 0.2\%$. Thus, for the vast majority of the spectrum ($j > 10$), $\theta_j$ is sufficiently small to make the sinc model analytically exact.

We rearrange the terms to match the form of the normalized sinc function, defined as $\text{sinc}(u) \triangleq \frac{\sin(\pi u)}{\pi u}$:

$$\lambda_j(B) \approx \left| \frac{\sin(\frac{B\theta_j}{2})}{\frac{B\theta_j}{2}} \right| \tag{21}$$

Let $\pi u = \frac{B\theta_j}{2}$, which implies $u = \frac{B\theta_j}{2\pi}$. Substituting $u$ yields the final approximation:

$$\lambda_j(B) \approx \left| \text{sinc}\left( \frac{B\theta_j}{2\pi} \right) \right| \tag{22}$$

This derivation confirms that mean pooling acts as a rectangular window filter in the signal domain, leading to the sinc-shaped spectral response shown in Figure 2.

### A.4. Relaxing the Locally Stationary Assumption

The locally stationary assumption on $c^{(j)}$ is used above only to isolate the effect of RoPE rotation and make the sinc attenuation factor transparent. The same conclusion still holds when the semantic content varies within a block. Let

$$c_k^{(j)} = \bar{c}^{(j)} + \delta_k^{(j)}, \qquad \bar{c}^{(j)} = \frac{1}{B} \sum_{k=0}^{B-1} c_k^{(j)}, \qquad \sum_{k=0}^{B-1} \delta_k^{(j)} = 0. \tag{23}$$

Then the pooled representation decomposes into a mean-content term and a residual variation term:

$$\bar{\mathbf{q}}^{(j)} = \frac{e^{in_0\theta_j}}{B} \left[ \bar{c}^{(j)} \sum_{k=0}^{B-1} e^{ik\theta_j} + \sum_{k=0}^{B-1} \delta_k^{(j)} e^{ik\theta_j} \right]. \tag{24}$$

The first term is exactly the component analyzed above and is attenuated by $\lambda_j(B)$. In the Dead Zone, where $\lambda_j(B) \approx 0$, this mean-content term is removed by destructive interference. The residual term

$$R_j = \frac{1}{B} \sum_{k=0}^{B-1} \delta_k^{(j)} e^{ik\theta_j} \tag{25}$$

is always bounded by

$$|R_j| \leq \frac{1}{B} \sum_{k=0}^{B-1} |\delta_k^{(j)}| \leq \max_k |\delta_k^{(j)}|, \tag{26}$$

so arbitrary intra-block variation cannot restore the vanished mean signal.

We can further obtain a tighter typical-case bound. Assume the variations have zero mean and per-token variance at most $\sigma_j^2$. Then

$$\mathbb{E}\left[ |R_j|^2 \right] = \frac{1}{B^2} \sum_{k,l} \mathbb{E}\left[ \delta_k^{(j)} \overline{\delta_l^{(j)}} \right] e^{i(k-l)\theta_j}. \tag{27}$$

For $k \neq l$, the cross terms are suppressed either because local semantic deviations are weakly correlated across positions, or because the rapidly rotating phase $e^{i(k-l)\theta_j}$ in the Dead Zone causes destructive interference over the sum. Keeping the diagonal terms gives

$$\mathbb{E}\left[ |R_j|^2 \right] \approx \frac{1}{B^2} \sum_{k=0}^{B-1} \mathbb{E}\left[ |\delta_k^{(j)}|^2 \right] \leq \frac{\sigma_j^2}{B}. \tag{28}$$

Thus the residual RMS scales as $\sigma_j/\sqrt{B}$. For the default $B = 128$, this yields a reduction factor $1/\sqrt{128} \approx 0.088$, consistent with the empirical energy collapse in Figure 3. The worst-case bound is tight only if the variations align adversarially as $\delta_k^{(j)} \propto e^{-ik\theta_j}$, which would require semantic content to oscillate at exactly the RoPE frequency and opposite phase across many Dead Zone dimensions. Such alignment is implausible in trained representations and is not observed empirically.

## B. Compatibility with YaRN

For Qwen3-8B, we use YaRN (Peng et al., 2024) to extend the native context length from 32K to 128K. For spectral-boundary analysis, the relevant component of YaRN is its NTK-by-parts interpolation, which modifies the RoPE frequency of each dimension as

$$\theta'_j = (1 - \gamma_j)\frac{\theta_j}{s} + \gamma_j \theta_j, \tag{29}$$

where $s$ is the extension ratio and $\gamma_j = \gamma(r_j)$. YaRN additionally applies an attention scaling factor, but that factor does not change the RoPE frequency boundaries analyzed here. The interpolation coefficient $\gamma_j$ is determined by $r_j = L\theta_j/(2\pi)$:

$$\gamma_j = \begin{cases} 1, & r_j > \beta, \\ 0, & r_j < \alpha, \\ \frac{r_j - \alpha}{\beta - \alpha}, & \alpha \leq r_j \leq \beta, \end{cases} \tag{30}$$

with default $\alpha = 1$ and $\beta = 32$. For Qwen3-8B, we have $b = 10^6$, $d = 128$, native context length $L = 32768$, and extension ratio $s = 4$. Computing $r_j$ over the spectral zones gives the following behavior:

| Zone | Dim. range | $r_j$ range | YaRN scaling | Effect on Prism |
|---|---|---|---|---|
| Dead | $2j \lesssim 30$ | $r_j \gtrsim 2.0 \times 10^2$ | $\gamma_j = 1$, unchanged | Attenuation analysis preserved |
| Transition (unchanged) | $30 \lesssim 2j \lesssim 47$ | $r_j > 32$ | $\gamma_j = 1$, unchanged | Unaffected |
| Transition (scaled) | $47 \lesssim 2j \lesssim 79$ | $1 < r_j < 32$ | Partial scaling, $\theta'_j < \theta_j$ | Attenuation reduced (favorable) |
| Semantic | $2j \gtrsim 79$ | $r_j < 1$ | Full scaling, $\theta'_j = \theta_j/s$ | Already near-lossless, remains so |

Therefore, YaRN does not modify the Dead Zone frequencies that motivate Prism's high-frequency branch. Within $d_{\text{high}} = 64$, dimensions up to $2j \approx 47$ are unchanged, and only the tail of the transition region receives mild scaling, which reduces rather than increases attenuation. The low-frequency branch covers the fully scaled long-wavelength region and the adjacent transition region; both are near-lossless under mean pooling or become less attenuated after frequency scaling. This explains why the same spectral split remains effective for Qwen3-8B after YaRN extension to 128K.

## C. Top-P Block Selection

Figure 10 provides the PyTorch-style implementation of the Top-P selection process used in Prism. The function takes block-level probabilities as input and sorts the key blocks for each query block based on relevance. Subsequently, it selects the minimal set of blocks required for the cumulative probability to exceed the threshold $p$. Finally, the original spatial order is restored via a scatter operation.

```python
def top_p(probs, p):
    # 1. Sort probabilities
    sorted_probs, sorted_indices = sort(probs, descending=True, dim=-1)

    # 2. Compute cumulative probabilities
    cumulative_probs = cumsum(sorted_probs, dim=-1)

    # 3. Thresholding
    sorted_mask = (cumulative_probs - sorted_probs) < p

    # 4. Scatter to restore order
    mask = zeros_like(probs)
    mask.scatter_(dim=-1, index=sorted_indices, src=sorted_mask)

    return mask
```

*Figure 10.* PyTorch-style implementation of the Top-P block selection.

# D. Experimental Setup Details

## D.1. Datasets

We provide detailed descriptions of the benchmarks used in our evaluation:

- **PG19 (Rae et al., 2020)**: A standard benchmark consisting of full-length books, used to evaluate the model's ability to model long-range dependencies via perplexity.

- **LongBench (Bai et al., 2024)**: A bilingual, multi-task benchmark consisting of 21 datasets across 6 task categories in both English and Chinese, designed to measure broader understanding capabilities.

- **RULER (Hsieh et al., 2024)**: A synthetic benchmark designed to measure the retrieval capability of long-context language models.

- **Video Benchmarks**: VideoMME (Fu et al., 2025) and LongVideoBench (Wu et al., 2024). We use max pixels of 327680 for each frame and 1 frame per second for video sampling, which translate to approximately 107K tokens per hour.

- **Video Generation**: We evaluate HunyuanVideo (Kong et al., 2024) with prompts sampled from VBench (Huang et al., 2024).

## D.2. HunyuanVideo Spectral Configuration

HunyuanVideo uses 3D-RoPE with RoPE base $b = 256$ and partitions each attention head into temporal, height, and width subspaces with dimensions $[16, 56, 56]$. Since the base is much smaller than that of LLM RoPE (e.g., $10^6$), high-frequency attenuation is stronger and the LLM setting $d_{\text{high}} = 64, d_{\text{low}} = 96$ is not directly transferable. We therefore apply the same attenuation criterion axis-wise. For an axis with subspace dimension $d_a$, the first cancellation point satisfies $B\theta_j \approx 2\pi$ with $\theta_j = b^{-2j/d_a}$, giving the real-dimension cutoff

$$2j_a^\star \approx d_a \cdot \frac{\ln(B/2\pi)}{\ln b}. \tag{31}$$

With $B = 128$ and $b = 256$, this yields $2j_t^\star \approx 8.7$ for the temporal subspace and $2j_h^\star = 2j_w^\star \approx 30.4$ for the spatial subspaces. Accordingly, we set the high-frequency branches to $d_{\text{high}}^t = 8$ and $d_{\text{high}}^h = d_{\text{high}}^w = 32$. For the low-frequency branches, we keep the tail dimensions while overlapping the transition region, using $d_{\text{low}}^t = 16$ and $d_{\text{low}}^h = d_{\text{low}}^w = 36$. This mirrors the overlap strategy used for LLM experiments: the high-frequency branch covers the strongly attenuated local-position dimensions, while the low-frequency branch preserves the more stable semantic and transition dimensions.

## D.3. Baselines Configuration

We compare Prism with the following baselines using their official implementations:

- **MInference**: A method employing offline search to classify attention heads into pre-defined heuristic patterns for subsequent block importance estimation. We use the recommended "Vertical-Slash" pattern configurations.

- **FlexPrefill**: An approach utilizing online search to dynamically switch between static patterns and mean-pooling based estimation depending on input contexts. We adopt $\gamma = 0.95, \tau = 0.1$ following the original paper.

- **XAttention**: A unified method introducing antidiagonal scoring to capture both geometric and semantic patterns without explicit head classification. We use threshold $p = 0.9$ and stride $S = 8$ following the original paper.

- **PBS-Attn**: A permutation-based block-sparse attention method that reorders tokens to cluster critical tokens, improving block-level sparsity for block selection. We use $p = 0.9$ and a segment size of 256 following the original paper.

We do not include SpargeAttention (Zhang et al., 2025) in the main quantitative comparison because it combines coarse-grained block estimation with orthogonal system optimizations such as Q/K quantization and warp-level block skipping. These optimizations are complementary to Prism's block-importance estimator and make it difficult to isolate the effect of the estimation mechanism itself. We therefore focus the main comparison on training-free dynamic sparse methods whose primary difference lies in how relevant blocks are selected.

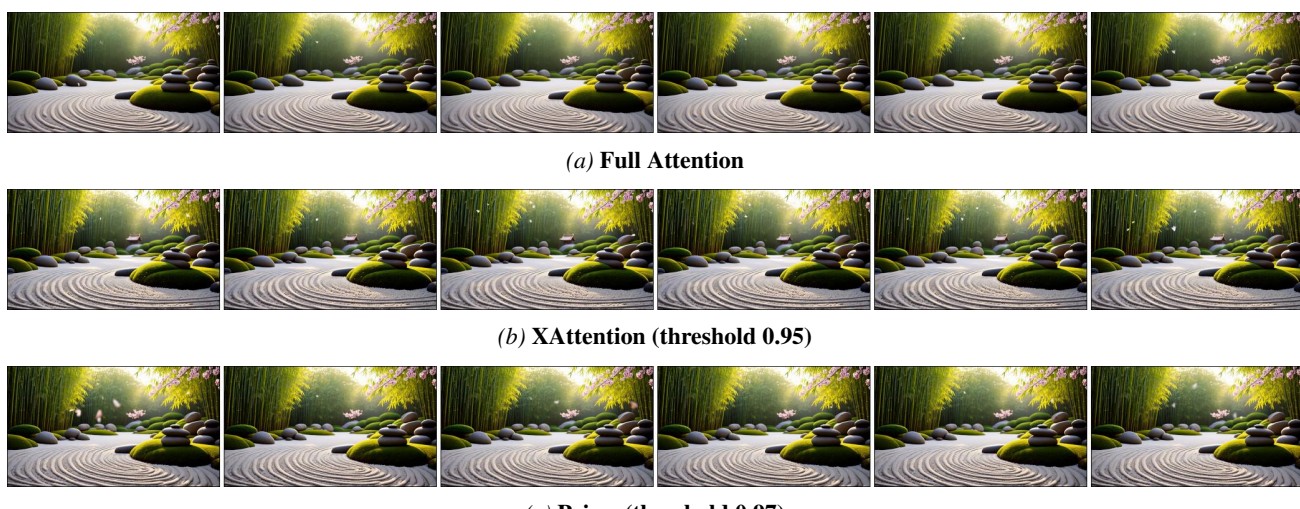

*(a)* **Full Attention**

*(b)* **XAttention (threshold 0.95)**

*(c)* **Prism (threshold 0.97)**

*Figure 11.* **Qualitative comparison on HunyuanVideo.** Each row shows generated frames from the same prompt under the corresponding attention implementation. XAttention uses threshold 0.95, and Prism uses threshold 0.97.

## E. Qualitative Video Generation Results

We provide qualitative HunyuanVideo results in Figure 11. The visual comparison uses the same prompt and sampling setup for full attention, XAttention, and Prism, showing that Prism preserves visual content and temporal consistency while improving efficiency.

## F. Effect of Block Size

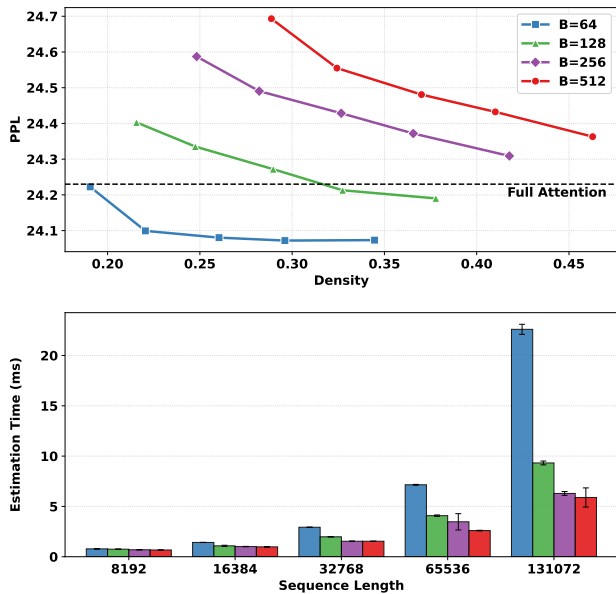

*Figure 12.* **Effect of Block Size** $B$**.** The upper panel illustrates the perplexity at various densities with a context length of 128K using Llama-3.1-8B-Instruct. The lower panels illustrates the estimation time at various sequence lengths.

Theoretically, a smaller block size $B$ enhances the Signal-to-Noise Ratio (SNR) by reducing spectral attenuation, but quadratically increases the estimation overhead due to the larger number of blocks ($N = L/B$). Figure 12 empirically validates this trade-off. In terms of accuracy (upper panel), finer granularity ($B = 64$) consistently yields better performance, even outperforming the full attention baseline due to effective noise filtering. $B = 128$ closely follows this trend, matching full attention at reasonable densities. However, in terms of efficiency (lower panel), the estimation latency for $B = 64$

rises sharply, reaching $\sim 22$ ms at 128K. Although this is still faster than many existing baselines (Figure 7), it is more than double the overhead of $B = 128$ ($\sim 9$ ms). Consequently, we select $B = 128$ for the main experiments, as a good compromise between accuracy and efficiency.

