# OpenReview forum: "Prism: Spectral-Aware Block-Sparse Attention"
_ICML.cc/2026/Conference — ICML 2026 regular_

### Official Review · Reviewer_pHfp · 2026-03-11

**Soundness:** 3
**Presentation:** 3
**Significance:** 2
**Originality:** 2
**Overall Recommendation:** 4
**Confidence:** 3

**Summary:**

This paper studies the problem of block-sparse attention for accelerating LLM pre-filling and finds that identifying relevant blocks efficiently remains a bottleneck. The authors trace the inaccuracy of standard coarse-grained attention via mean pooling to the interaction between mean pooling and Rotary Positional Embeddings (RoPE) and prove that mean pooling acts as a low-pass filter that induces destructive interference in high-frequency dimensions. To address this issue, the authors propose a training-free spectral-aware approach that decomposes block selection into high-frequency and low-frequency branches. Empirical results validate that the proposed method maintains the accuracy on various benchmarks and brings significant speedup in long-context scenarios.

**Compliance With Llm Reviewing Policy:**

Affirmed.

**Final Justification:**

Thanks for the detailed responses and additional experiments provided by the authors. They have addressed my concerns, and I tend to maintain the rating.

**Key Questions For Authors:**

Please refer to weakness 1,2,3.

**Limitations:**

Yes

**Strengths And Weaknesses:**

Strength:
1. The authors trace the inaccuracy of standard coarse-grained attention via mean pooling to the interaction between mean pooling and Rotary Positional Embeddings (RoPE) and may provide insights into future work in this area.
2. Experiments are extensive. The authors have conducted experiments on different benchmarks, tasks, and models to demonstrate that the proposed training-free method maintains performance with full attention in various cases.
3. The presentation seems good. Overall, the paper is well-organized and the writing is clear.

Weakness:
1. The reviewer observes that there is some misalignment with the performance of the baseline methods. For FlexPrefill and XAttention, both papers have shown that they could outperform full attention, while the results in this paper show that they were worse than full attention on the same benchmark. It would be helpful to explain the phenomenon.
2. Moreover, it would be beneficial to extend the method to dense prediction tasks like video generation, following XAttention. It would strengthen the paper if the method could generalize to dense prediction tasks as well.
3. Further, the inference speedup results in Fig. 5 and Fig. 6 seem to be quite different from the results in XAttention (Fig. 4). A detailed explanation of the inference setup could be helpful to better understand the method.

---

> ### Author Rebuttal · Authors · 2026-03-30
>
> We thank the reviewer for the thorough review. We address each concern below.
>
> **Response to W1:**
>
> We thank the reviewer for the question. Evaluation results can vary based on hardware setups, evaluation frameworks, prompt formatting, etc. **Minor differences in these factors can lead to variation in absolute scores; what matters is the relative comparison under identical conditions.** As stated in Appendix C.2, **we use the recommended configurations for all baselines with their official implementations, and all methods are evaluated under the same framework for fair comparison.** More importantly, all our implementations (including baseline integrations and the evaluation pipeline) are **open-sourced in the anonymous repo** linked in the paper abstract, providing one-liner scripts for easy reproduction. In all honesty, one exception is XAttention's offline-searched per-layer per-head thresholds, which are not used in our comparison since: 1. the code for threshold searching is not open-sourced; 2. they only released the searched results for Llama, which cannot be directly ported to other models.
>
> **Response to W2:**
>
> Thank you for pointing this out. Prism indeed naturally extends to video generation. We provide both theoretical analysis and empirical validation:
>
> Video DiTs typically use 3D-RoPE, where $d$ is factorized into orthogonal segments $d_t, d_h, d_w$. Since the rotations are orthogonal, mean pooling decouples into independent 1D attenuations per axis, and Eq. 6 applies directly:
> $$\lambda_{j, t}(B_t) \approx \text{sinc}\left(\frac{B_t \theta_j}{2\pi}\right)$$
>
> More importantly, 3D-RoPE is a special case of M-RoPE for video tokens used in QwenVL, which we have shown to be compatible with Prism empirically (Tab. 3).
>
> Empirically, we verify this by applying Prism to HunyuanVideo, evaluating on VBench, same settings as in the XAttention paper. Due to the time constraint, we sample 100 out of 946 prompts for evaluation. Note that, for 3D-RoPE in HunyuanVideo, the RoPE base $b$ is 256, which is significantly lower than in LLMs, **making the destructive interference much more severe**. The head dimension $d$ is 128, with layout of [16, 56, 56] for temporal, height, and width, respectively. Following Eq. 8 in our paper, we thus select $d_{low_t}=16, d_{high_t}=8$, $d_{low_h}=d_{low_w}=36, d_{high_h}=d_{high_w}=32$. For XAttention, we use $S$=8, threshold $\tau=0.9/0.95$, 5 warmup steps, identical to the original paper.
>
> |Method($\tau$)|PSNR↑|SSIM↑|LPIPS↓|Speedup|
> |---|---|---|---|---|
> |XAttn(0.9)|22.25|0.7567|0.2005|1.54x|
> |XAttn(0.95)|24.34|0.8215|0.1476|1.33x|
> |Prism(0.9)|21.47|0.7163|0.2346|1.78x|
> |Prism(0.93)|22.39|0.7572|0.2001|1.62x|
> |Prism(0.95)|23.14|0.7797|0.1809|1.5x|
> |Prism(0.97)|24.37|0.8247|0.1445|1.37x|
>
> **The VBench results show that at comparable quality, Prism achieves higher speedups than XAttention, consistent with our results in the paper.**
>
> **We also provide qualitative results at https://anonymous.4open.science/r/prism_anonymous-7E36/video_qualitative_results/**, which contains generated videos from FlashAttention, XAttention($\tau=0.95$), and Prism($\tau=0.97$) with the same prompt. The results show that the generation of Prism is more consistent with full attention.
>
> **Response to W3:**
>
> There are a few sources that could cause this discrepancy:
>
> 1. The baseline full attention implementation is different. The XAttention paper uses the FlashInfer backend, whereas we use PyTorch SDPA with the FlashAttention-2 backend.
> 2. XAttention did not explicitly disclose the specific GPU type used for evaluation. We used H100 for efficiency evaluation, so the hardware setup could be different.
> 3. The original XAttention results are likely evaluated with offline-searched thresholds, which could bring extra benefits but are non-trivial to generalize to other models (as we discussed in W1).
>
> **Besides these setup discrepancies, our results remain valid based on a complexity analysis of estimation overhead.** XAttention samples $S$ antidiagonal elements per $S\times S$ grid, yielding $\mathcal{O}(L^2d/S)$. Prism compresses $B$ tokens into one block-level representative for a dense matmul, yielding $\mathcal{O}(L^2d/B^2)$. With default parameters ($S=8$, $B=128$):
>
> ||XAttention|Prism|
> |---|---|---|
> |Estimation complexity|$\mathcal{O}(L^2d/S)$|$\mathcal{O}(L^2d/B^2)$|
> |Reduction vs. Full Attn|$8\times$ ($S$=8)|$16{,}384\times$ ($B$=128)|
> |Memory access|Strided (non-contiguous)|Dense matmul (Tensor Core friendly)|
>
> This $2048\times$ gap manifests in practice: at 128K, XAttention's estimation reaches ~85ms (Fig. 7), capping speedup at ~$3\times$; Prism's ~9ms enables $5.1\times$ (Fig. 6). **This gap widens with $L$, as Prism's estimation cost stays constant at $1/B^2\approx0.006\%$ of full attention regardless of sequence length.** Furthermore, Prism's spectral bands depend only on the RoPE base $b$ and head dimension $d$ (Eq. 8), not on $L$, so calibration generalizes to arbitrary context lengths.

---

> > ### Author Rebuttal · Reviewer_pHfp · 2026-04-04
> >
> > Thanks for the detailed responses and additional experiments provided by the authors. They have addressed my concerns, and I tend to maintain the rating.

---

> > > ### Author Response · Authors · 2026-04-07
> > >
> > > Thank you for the acknowledgement, and we are glad that our rebuttal addresses all the concerns. We especially appreciate the suggestion on video generation, which directly led to the HunyuanVideo experiments that we believe substantially strengthen the paper's evaluation breadth.
> > >
> > > For the reviewer's convenience, we summarize the key improvements made during the rebuttal period across all reviewer threads:
> > >
> > > 1. **New experiment: video generation on HunyuanVideo** (Response to Reviewer pHfp W2). We applied Prism to HunyuanVideo with 3D-RoPE ($b$=256), demonstrating comparable quality to XAttention with higher speedup on VBench. Qualitative video results are also provided.
> > > 2. **New baseline: SpargeAttention comparison** (Response to Reviewer T9cg W4). Prism achieves higher accuracy (41.08 vs 40.47 on LongBench) with 4.3$\times$ better speedup at 128K.
> > > 3. **Tighter theoretical analysis** (Follow-up to Reviewer T9cg W1). We relaxed the locally-stationary assumption on $c^{(j)}$ and derived a $\sigma/\sqrt{B}$ probabilistic bound on the residual under general intra-block content variation, closely matching the empirical $10\times$ energy collapse in Fig. 3 ($1/\sqrt{128} \approx 0.088$).
> > > 4. **YaRN compatibility analysis** (Response to Reviewer T9cg W2). Zone-by-zone analysis showing Prism's spectral bands remain valid under YaRN frequency scaling.
> > > 5. **Estimation complexity analysis** (Response to Reviewer T9cg W3 / Reviewer pHfp W3). Prism's block-level estimation yields $\mathcal{O}(L^2d/B^2)$, a $2048\times$ reduction over XAttention's $\mathcal{O}(L^2d/S)$, with estimation cost staying constant at $1/B^2 \approx 0.006\%$ of full attention regardless of sequence length.
> > > 6. **Decoding applicability analysis** (Response to Reviewer p3ay Limitations). Concrete pipeline sketch for extending Prism to autoregressive decoding.
> > >
> > > We believe these improvements, including new experiments, new baselines, and strengthened theory, collectively address all identified limitations. We kindly ask the reviewer to consider the full picture when finalizing the assessment. Thank you for your time and effort throughout the review process.

---

### Official Review · Reviewer_T9cg · 2026-03-12

**Soundness:** 3
**Presentation:** 3
**Significance:** 3
**Originality:** 3
**Overall Recommendation:** 4
**Confidence:** 4

**Summary:**

This paper introduces a training-free spectral-aware approach for block-sparse attention in long-context LLM prefilling. The authors claim that the weakness of standard coarse-grained attention, based on mean-pooled block representations, can be attributed to a spectral interaction with RoPE: mean pooling behaves like a low-pass filter, attenuating high-frequency positional components and thereby obscuring local, slash-like attention structures. Based on this assumption, the paper proposes the method named PRISM, which separates block selection into high-frequency and low-frequency branches, applies energy/RMS-based temperature calibration to restore the attenuated signals, and performs block selection using only block-level operations. The authors experimented in various benchmark datasets and achieved fast attention while maintaining performance.

**Compliance With Llm Reviewing Policy:**

Affirmed.

**Final Justification:**

In light of the rebuttal, I raise my Soundness score to 3 and my Overall Recommendation to 4.

**Key Questions For Authors:**

1. Locally stationary assumption: The derivation assumes c^(j) is constant within a pooling window, and Prism provides no token-level fallback. Could the authors characterize the performance impact when this assumption is failed, e.g., on code or multilingual sequences?
2. YaRN interaction with spectral bands: Were spectral bands re-examined after applying YaRN to Qwen3-8B? If not, please explain why the spectral boundary mismatch between the original base b=10^6 and the YaRN-modified frequency distribution does not affect estimation quality.
3. SpargeAttention exclusion: SpargeAttention is cited in Related Work in the same design space as Prism, yet absent from quantitative comparisons. Could you clarify the reason for its exclusion?

**Limitations:**

Partially. The following points may discussion: the locally stationary content assumption underlying the theoretical framework is not acknowledged as a practical limitation; the interaction between fixed spectral band boundaries and RoPE variants such as YaRN is not addressed.

**Strengths And Weaknesses:**

Strengths.

1. Theoretical foundation with empirical validation: This paper presents an analysis of how mean pooling under RoPE causes frequency-dependent attenuation, creating blind spots for high-frequency positional information. Attenuation analysis, dead/transition/semantic zone analysis, and RMS comparison are presented, along with a mathematical proof, providing a relatively persuasive explanation of why conventional sparse attention loses local positional information after pooling.
2. Hyperparameter tuning-free calibration: The energy-based temperature calibration is derived from the spectral energy distribution of pooled matrices, requiring no additional tuning. This is a meaningful practical advantage.
3. Low overhead due to block-level operations: Unlike XAttention, which uses antidiagonal token-level operations, and MInference/FlexPrefill, which involve offline/online token-level search, Prism's entire pipeline consists of matrix multiplications over pooled block representations. (Fig.7 directly demonstrates the consequence)

Weaknesses.

1. About c^(j): The authors assume that the semantic content is locally stationary within the pooling window. This assumption may not hold for heterogeneous sequences such as code or multilingual text. Unlike prior methods (MInference, FlexPrefill) that retain token-level verification as a fallback, Prism completely eliminates token-level estimation. The paper does not provide a clean theoretical basis for this assumption or an analysis of its failure. The RMS analysis is useful supporting evidence, but it is still only a partial validation of the full causal story.
2. About implementation detail d_high=64 and d_low=96: The authors set the spectral bands to d_high=64,d_low=96 and did not change them. Qwen3-8B's native context is 32K, YaRN is applied to extend it to 128K. YaRN may modifies the effective RoPE frequency scaling, which directly alters the spectral attenuation profile described by Eq.6 to Eq.8 and shifts the zone boundaries. The spectral bands were derived from an analysis assuming the original base b=10^6, and the paper does not discuss whether these remain optimal after YaRN modification. This is not discussed in the main paper, but the authors said "generalizability to RoPE variants without requiring additional adaptations.” in the main results.
3. About scalability claim: The paper presents Prism as a “robust and scalable solution for long-context and multimodal LLMs”, yet evaluation results is capped at 128K. To substantiate the scalability claim, a brief analysis of how estimation overhead evolves beyond 128K, such as a theoretical projection or latency extrapolation, would be valuable.
4. About missing baseline: SpargeAttention is introduced in the Related Work as a coarse-level estimation method targeting all attention heads, which places it in the same design space as Prism. However, it is excluded from all comparisons without explicit justification.

---

> ### Author Rebuttal · Authors · 2026-03-30
>
> We thank the reviewer for the thorough evaluation. We address each concern below.
>
> **Response to W1/Q1:**
>
> **The locally-stationary assumption on $c^{(j)}$ is introduced solely to isolate the effect of RoPE rotation and make the derivation of the sinc attenuation factor (Eq. 6) more transparent, not as a prerequisite for our conclusion.** Consider the general case where $c^{(j)}$ varies within a block. Let $c_k^{(j)} = \bar{c}^{(j)} + \delta_k^{(j)}$ with $\bar{c}^{(j)}=\frac{1}{B}\sum_k c_k^{(j)}$ and $\sum_k\delta_k^{(j)}=0$. The mean-pooled vector decomposes as:
>
> $$\bar{q}^{(j)}=\frac{e^{in_0\theta_j}}{B} \left[ \bar{c}^{(j)} \cdot \sum_{k=0}^{B-1} e^{ik\theta_j} + \sum_{k=0}^{B-1} \delta_k^{(j)} e^{ik\theta_j} \right]$$
>
> In the Dead Zone($\lambda_j \approx 0$), this first term vanishes due to destructive interference of RoPE rotations. The residual $\frac{1}{B}\sum_k \delta_k^{(j)} e^{ik\theta_j}$ is bounded by $\max_k |\delta_k^{(j)}|$. Fig. 3 empirically confirms this: Dead Zone energy collapses to RMS $\approx 0.1$ on real representations **without any assumptions. High-freq dimensions are wiped regardless of intra-block content variation.** Prism's spectral separation and calibration address this signal loss, validated by ablations (Figs. 8-9) and benchmarks (Tabs. 1-3).
>
> Regarding the token-level fallback: Prism's exclusively block-level pipeline is a design advantage, enabling superior efficiency scaling (Fig. 7). Accuracy is maintained even on token-sensitive tasks like Code (Prism 18.72 vs. Full 18.01) and RULER (Tab. 2). **These results demonstrate that block-level operations with proper spectral calibration are sufficient for accurate block importance estimation.**
>
> **Response to W2/Q2:**
>
> YaRN modifies each dimension's frequency as $\theta_j'=(1-\gamma_j)\cdot\theta_j/s+\gamma_j\cdot\theta_j$, where $s$ is the extension ratio and $\gamma_j$ is determined by the ratio $r_j=L\theta_j/(2\pi)$: $\gamma_j=1$ (no scaling) when $r_j>\beta$, $\gamma_j=0$ (full scaling) when $r_j<\alpha$, with a linear ramp in between (default $\alpha=1, \beta=32$).
>
> For Qwen3-8B ($b=10^6$, $d=128$, $L=32768$, $s=4$), we compute $r_j$ for each spectral zone:
>
> |Zone|Dim range|$r_j$ range|YaRN scaling|Effect on Prism|
> |---|---|---|---|---|
> |Dead|$2j \lesssim 30$|$r_j \gg 32$|$\gamma_j=1$, **unchanged**|Attenuation analysis preserved|
> |Transition (1st half)|$30 \lesssim 2j \lesssim 40$|$r_j>32$|$\gamma_j=1$, **unchanged**|Unaffected|
> |Transition (2nd half)|$40 \lesssim 2j \lesssim 60$|$1<r_j<32$|Partial, $\theta_j$ reduced|Attenuation **reduced** (favorable)|
> |Semantic|$2j \gtrsim 60$|$r_j<1$|Full scaling, $\theta_j'=\theta_j/s$|Already near-lossless, remains so|
>
> **YaRN does not modify Dead Zone frequencies.** Within $d_{\text{high}}=64$, only the tail ($2j \in [40,64]$) receives mild scaling that **reduces** attenuation. The Semantic Zone becomes **even more lossless**. Qwen3-8B results (Tabs. 1-3) validate this: unchanged bands work well after YaRN extension to 128K.
>
> **Response to W3:**
>
> Prism's scalability stems from its pure block-level estimation. Taking XAttention as a representative example, it samples $S$ antidiagonal tokens per $S\times S$ grid, yielding $\mathcal{O}(L^2d/S)$. Prism compresses $B$ tokens into one block-level representative and performs a dense matmul, yielding $\mathcal{O}(L^2d/B^2)$.
> With default parameters ($S=8$, $B=128$):
> ||XAttention|Prism|
> |---|---|---|
> |Estimation complexity|$\mathcal{O}(L^2d/S)$|$\mathcal{O}(L^2d/B^2)$|
> |Reduction vs. Full Attn|$8\times$ ($S$=8)|$16{,}384\times$ ($B$=128)|
> |Memory access|Strided (non-contiguous)|Dense matmul (Tensor Core friendly)|
>
> This $2048\times$ theoretical gap manifests in practice: at 128K, XAttention's estimation reaches ~85 ms (Fig. 7), capping end-to-end speedup at ~$3\times$; Prism's ~9ms enables $5.1\times$ (Fig. 6). **This gap widens with $L$**.
>
> Prism's spectral band configuration depends only on $b$ and $d$ (Eq. 8), generalizing to arbitrary $L$. **Its relative estimation cost remains a constant $1/B^2\approx0.006%$, staying negligible across any sequence length.**
>
> **Response to W4/Q3:**
>
> We exclude SpargeAttn because: 1. **It bundles orthogonal optimizations (Q/K quantization, warp-level block skipping) beyond block estimation.**; 2. **Its performance-efficiency trade-off is less competitive than its concurrent work, XAttention.** We select XAttention as it solely focuses on block estimation, matching Prism's scope.
> We provide SpargeAttn results using official kernels with recommended hyperparameters ($\tau=0.6$, $\theta=0.98$, $\lambda=50$) on Llama3.1-8B:
>
> LongBench:
> |Method|Single-Doc QA|Multi-Doc QA|Summarization|Few-shot Learning|Code|Synthetic|Avg.|
> |---|---|---|---|---|---|---|---|
> |SpargeAttn|45.58|42.31|26.03|46.42|18.01|64.49|40.47|
> |Prism|47.09|42.13|26|46.4|18.72|66.15|41.08|
>
> Speedup:
> |Method|8K|16K|32K|64K|128K|
> |---|---|---|---|---|---|
> |SpargeAttn|0.76x|0.83x|0.92x|1.13x|1.19x|
> |Prism|1.21x|1.75x|2.66x|3.80x|5.15x|

---

> > ### Author Rebuttal · Reviewer_T9cg · 2026-04-03
> >
> > Thank you for the thorough rebuttal. The responses to W2, 3, 4 are well-taken and satisfactorily address those concerns. However, one issue remains partially unresolved.
> > On W1, while the authors show that the Dead Zone collapse holds regardless of intra-block content variation, the residual term is bounded by a loose max∣δk​∣ rather than a tighter average-based or probabilistic bound.
> >
> > After Comment,
> > The authors' rebuttal has addressed most of my concerns. I would appreciate it if you could review the points that might cause confusion and incorporate them into the revised version of the paper. I will raise overall recommendation.

---

> > > ### Author Response · Authors · 2026-04-03
> > >
> > > We sincerely thank the reviewer for the careful follow-up, the excellent mathematical intuition, and for confirming that W2–W4 are resolved!
> > >
> > > **We originally used the $\max_k|\delta_k^{(j)}|$ bound deliberately because it is assumption-free (holding unconditionally via the triangle inequality) and was sufficient to establish that the original clean signal is destroyed.** We chose to let the empirical validation (Fig. 3) speak to the actual magnitude of the residual rather than introduce distributional assumptions. That said, we fully agree that the worst-case bound is loose, and we are happy to provide a tighter probabilistic analysis that perfectly explains our empirical observations.
> > >
> > > **Tighter Probabilistic Bound**
> > > Let the residual be $R = \frac{1}{B}\sum_k \delta_k^{(j)} e^{ik\theta_j}$. We compute its expected squared magnitude (variance):
> > > $$\mathbb{E}\left[|R|^2\right] = \frac{1}{B^2}\sum_{k,l} \mathbb{E}[\delta_k^{(j)}\overline{\delta_l^{(j)}}] \cdot e^{i(k-l)\theta_j}$$
> > >
> > > The cross-terms ($k \neq l$) are heavily suppressed by two mechanisms:
> > > 1. If the variations $\delta_k^{(j)}$ are approximately uncorrelated across positions, then $\mathbb{E}[\delta_k\overline{\delta_l}] \approx 0$.
> > > 2. Even under local correlation, the rapid phase rotation $e^{i(k-l)\theta_j}$ in the Dead Zone causes destructive interference over the sum—the exact same geometric mechanism that kills the mean term.
> > >
> > > Either effect suppresses the cross-terms, leaving only the diagonal ($k=l$):
> > > $$\mathbb{E}\left[|R|^2\right] \approx \frac{1}{B^2}\sum_{k=0}^{B-1} \mathbb{E}[|\delta_k^{(j)}|^2] = \frac{\sigma^2}{B}$$
> > >
> > > Taking the square root, the expected magnitude (RMS) of the residual scales as **$\sigma/\sqrt{B}$**, giving an $\sim 11\times$ reduction from the per-token deviation scale for $B=128$ ($1/\sqrt{128} \approx 0.088$).
> > >
> > > **When is the worst-case bound tight?**
> > > The loose $\max$ bound saturates *only* if all terms $\delta_k^{(j)} e^{ik\theta_j}$ align in the complex plane. This requires $\delta_k^{(j)} \propto e^{-ik\theta_j}$—meaning the model's semantic content deviations would need to encode a sinusoidal signal at exactly the RoPE frequency with opposite phase, perfectly canceling the rotation. This is highly implausible because:
> > > 1. There is no training signal forcing learned semantic representations to oscillate at fixed RoPE frequencies.
> > > 2. It would need to happen simultaneously across all Dead Zone dimensions (each with a different $\theta_j$).
> > > 3. If such adversarial alignment existed, Fig. 3 would show high residual energy in specific layers/heads. Instead, we see a uniform energy collapse.
> > >
> > > **Empirical Confirmation**
> > > This probabilistic bound perfectly explains our data. Fig. 3 shows the token-level Dead Zone energy (RMS $\approx 1.0$) collapsing to RMS $\approx 0.1$ after pooling. **This $10\times$ reduction closely matches the theoretical expectation of $1/\sqrt{128} \approx 0.088$. This confirms the residual is governed by $\mathcal{O}(1/\sqrt{B})$ statistical concentration rather than worst-case behavior.**
> > >
> > > Thank you again for the feedback, as it makes the theoretical framework significantly tighter. We will incorporate this variance analysis in the revision.
> > >
> > > We hope that the resolution of W2–W4 (as acknowledged by the reviewer) together with the tighter probabilistic analysis for W1 address all remaining concerns, and we kindly ask the reviewer to consider updating the score accordingly. Thanks again for your time and effort.

---

### Official Review · Reviewer_p3ay · 2026-03-14

**Soundness:** 2
**Presentation:** 3
**Significance:** 3
**Originality:** 3
**Overall Recommendation:** 4
**Confidence:** 5

**Summary:**

The paper identifies a theoretical limitation of mean-pooled coarse-grained attention under RoPE, showing that mean pooling acts as a low-pass filter that suppresses high-frequency positional signals and leads to inaccurate block importance estimation. To address this, it proposes Prism, a training-free spectral-aware method that restores positional signals via frequency-aware calibration, enabling efficient block-level selection and achieving up to 5.1× speedup while maintaining accuracy close to full attention.

**Compliance With Llm Reviewing Policy:**

Affirmed.

**Key Questions For Authors:**

In weakness.

**Limitations:**

No. The discussion of limitations is limited. In particular, the paper focuses on the pre-filling stage, and it is unclear whether the proposed method would also work effectively during decoding. A brief discussion on this aspect would improve the paper.

**Strengths And Weaknesses:**

Strength 1: The paper provides a detailed analysis of the interaction between RoPE and block-wise sparse attention, and offers a clear explanation of why standard coarse-grained estimation can fail.

Strength 2: The proposed method introduces separate high-frequency and low-frequency estimations for Q and K, together with energy-based calibration, which is a well-motivated design.

Weakness 1: SeerAttention uses max pooling and min pooling for K and mean pooling for Q. Their ablation suggests this works well, with the explanation that K contains more high-frequency information while Q is more dominated by low-frequency information. It would be helpful to compare with SeerAttention and discuss whether your spectral analysis is consistent with, or provides additional insight into, that design choice.

Weakness 2: The improvement on RULER is not very significant. On this retrieval benchmark, the results seem to mainly show that the method does not hurt performance, rather than demonstrating a clear advantage over prior approaches. It would be helpful if the paper could better explain why the gains are limited on retrieval-heavy tasks.

[1] SeerAttention: Learning Intrinsic Sparse Attention in Your LLMs, arXiv:2410.13276, NeurIPS 2025

---

> ### Author Rebuttal · Authors · 2026-03-30
>
> We sincerely thank the reviewer for recognizing the detailed theoretical analysis of the RoPE–pooling interaction (Strength 1) and the well-motivated design of frequency-decomposed estimation with energy-based calibration (Strength 2). We are glad that the reviewer finds our spectral analysis provides a clear explanation for why standard coarse-grained estimation fails. We address reviewer's questions below.
>
> **Response to W1: Comparison with SeerAttention and spectral consistency.**
>
> We appreciate this insightful connection. We’d like to clarify that **Prism and SeerAttention operate in fundamentally different design spaces**. SeerAttention applies pooling to **pre-RoPE** **representations** and processes them through **learned** gates trained via self-distillation, while Prism operates on **post-RoPE** **representations** using **training-free** spectral calibration. **As a result, their design choices are not directly transferable to the post-RoPE setting.** In our pilot experiments, we tested max/min pooling for K for coarse-grained attention in our  (training-free+post-RoPE) and observed significant performance degradation in retrieval-heavy tasks like RULER (the “needle” to be retrieved does not necessarily have the largest magnitude for all its dimensions), which motivated our use of mean pooling as the foundation for the subsequent research.
>
> **Notably, the Q/K pooling asymmetry SeerAttention observes is attributed to outlier distribution differences in K (consistent with findings in the LLM quantization literature), rather than to frequency-domain effects introduced by RoPE.** Prism's spectral analysis reveals a **separate and previously unidentified failure mode** — destructive interference from averaging over RoPE-rotated vectors — that outlier-aware pooling strategies alone would not resolve. Furthermore, operating on post-RoPE representations is the more common and practical approach for training-free methods, as it avoids interfering with the model's original computation flow; this is the same paradigm adopted by MInference, FlexPrefill, XAttention, etc. Regarding empirical comparison, we note that more recent training-free baselines (e.g., XAttention) have reported stronger results than SeerAttention, which motivated our baseline selection. We will discuss SeerAttention and clarify this distinction of pooling methods in the revised paper.
>
> **Response to W2: Limited improvement on RULER.**
>
> We agree that RULER results primarily demonstrate parity across methods, yet we argue this is consistent with our spectral analysis. RULER tasks are predominantly semantic retrieval tasks where the query must locate specific "needles" in context. This aligns with the behavior of the low-frequency band (analyzed in Sec. 3.1 and Fig. 1), which is well-preserved by mean pooling ($\lambda_j \approx 1$ in the Semantic Zone, Fig. 2). Since all methods already perform well in this regime, the room for improvement from high-frequency correction is inherently limited.
>
> Nevertheless, we highlight two points: (1) **Prism achieves this parity using exclusively block-level operations**, whereas MInference and FlexPrefill rely on token-level estimation with the last query block (a heuristic that is inherently advantaged on RULER's format where the query is positioned at the end) resulting in substantially higher estimation overhead (Fig. 7); (2) **Prism's advantage surfaces clearly on tasks that exercise local positional coherence** (high-frequency slash patterns), including language modeling (PG19, $\Delta PPL \approx 0$ vs. significant degradation for baselines), long-context understanding (LongBench), and video understanding (VideoMME Long, where Prism surpasses even full attention at 64.00 vs. 63.11). These results suggest that Prism's spectral calibration is most impactful precisely where existing methods struggle, preserving fine-grained positional structure, while remaining competitive on retrieval tasks where this structure is less critical.
>
> **Response to Limitations: Applicability to decoding.**
>
> We thank the reviewer for this thoughtful suggestion. Prism is currently designed for the prefilling stage, where the full sequence is available for block-level pooling. During autoregressive decoding, the query is a single token, so query-side pooling is unnecessary; the query can be directly split into high-frequency and low-frequency components and scored against the spectrally decomposed pooled key caches ($\bar{K}\_{\text{high}}$ and $\bar{K}\_{\text{low}}$, which are already computed during prefill and can be incrementally updated). This actually simplifies Prism's pipeline while preserving the core spectral calibration on the key side, where the attenuation analysis (Eq. 6–7) still fully applies. We leave the implementation of this for future work and will include this discussion in the revised paper.

---

> > ### Author Rebuttal · Reviewer_p3ay · 2026-04-04
> >
> > Thanks! I will maintain my score as week accept!

---

> > > ### Author Response · Authors · 2026-04-07
> > >
> > > Thank you for the acknowledgement, and we are glad that our rebuttal addresses all the concerns. We also appreciate the insightful suggestion on SeerAttention, which helped us better articulate the distinction between pre-RoPE and post-RoPE design spaces.
> > >
> > > For the reviewer's convenience, we summarize the key improvements made during the rebuttal period across all reviewer threads:
> > >
> > > 1. **New experiment: video generation on HunyuanVideo** (Response to Reviewer pHfp W2). We applied Prism to HunyuanVideo with 3D-RoPE ($b$=256), demonstrating comparable quality to XAttention with higher speedup on VBench. Qualitative video results are also provided.
> > > 2. **New baseline: SpargeAttention comparison** (Response to Reviewer T9cg W4). Prism achieves higher accuracy (41.08 vs 40.47 on LongBench) with 4.3$\times$ better speedup at 128K.
> > > 3. **Tighter theoretical analysis** (Follow-up to Reviewer T9cg W1). We relaxed the locally-stationary assumption on $c^{(j)}$ and derived a $\sigma/\sqrt{B}$ probabilistic bound on the residual under general intra-block content variation, closely matching the empirical $10\times$ energy collapse in Fig. 3 ($1/\sqrt{128} \approx 0.088$).
> > > 4. **YaRN compatibility analysis** (Response to Reviewer T9cg W2). Zone-by-zone analysis showing Prism's spectral bands remain valid under YaRN frequency scaling.
> > > 5. **Estimation complexity analysis** (Response to Reviewer T9cg W3 / Reviewer pHfp W3). Prism's block-level estimation yields $\mathcal{O}(L^2d/B^2)$, a $2048\times$ reduction over XAttention's $\mathcal{O}(L^2d/S)$, with estimation cost staying constant at $1/B^2 \approx 0.006\%$ of full attention regardless of sequence length.
> > > 6. **Decoding applicability analysis** (Response to Reviewer p3ay Limitations). Concrete pipeline sketch for extending Prism to autoregressive decoding.
> > >
> > > We believe these improvements, including new experiments, new baselines, and strengthened theory, collectively address all identified limitations. We kindly ask the reviewer to consider the full picture when finalizing the assessment. Thank you for your time and effort throughout the review process.

---

### Decision · Program_Chairs · 2026-04-30

**Decision:**

Accept (regular)

**Comment:**

This paper presents a training-free, spectral-aware block-sparse attention method that speeds up long-context LLM pre-filling by addressing high-frequency interference caused by mean pooling on RoPE embeddings.

All reviewers recommended acceptance (4, 4, 4), appreciating the thorough theoretical analysis.
Initial concerns included the reliance on a locally stationary assumption, missing comparisons to related methods, unclear applicability to video generation. In the rebuttal, the authors resolved all these concerns thoughtfully.

With strong theoretical justification and empirical confirmation, AC recommends Accept.